# Reduced cytochrome P-450 (CYP) 2D6 activity and *Plasmodium vivax* malaria risk in Amazonians: A retrospective, population-based cohort study

Rodrigo M. Corder[1,2,3]*, Maria Carolina S. B. Puça[4,5◉], Carlos A. Prete[1,6], Winni A. Ladeia[1],
Priscila T. Rodrigues[1,7], Igor C. Johansen[1,8], Tais Nobrega de Sousa[4,5‡],
Marcelo U. Ferreira [1,3]*, on behalf of the Mâncio Lima Cohort Study Working Group[¶]

1 Department of Parasitology, Institute of Biomedical Sciences, University of São Paulo, São Paulo, Brazil,
2 Department of Life Sciences, Faculty of Natural Sciences, Imperial College London, London, United
Kingdom, 3 Global Health and Tropical Medicine (GHTM), Institute of Hygiene and Tropical Medicine,
Nova University of Lisbon, Lisbon, Portugal, 4 Molecular Biology and Malaria Immunology Research
Group, René Rachou Institute, Fiocruz, Belo Horizonte, Brazil, 5 Department of Microbiology, Tumour and
Cell Biology, Karolinska Institute, Solna, Sweden, 6 Department of Communications, School of Electrical
and Computer Engineering, State University of Campinas, Campinas, Brazil, 7 Brazilian Biorenewables
National Laboratory, Brazilian Center for Research in Energy and Materials, Campinas, Brazil,
8 Department of Demography, Institute of Philosophy and Humanities, State University of Campinas,
Campinas, Brazil

¶ Membership of the Mâncio Lima Cohort Study Working Group is listed in the Acknowledgments.
◉ Contributed equally as joint first authors.
‡ Contributed equally as joint senior authors.
* rodrigo.corder@usp.br (RMC); muferrei@usp.br (MUF)

pntd.0014160

Witwatersrand Johannesburg, SOUTH AFRICA

**Peer Review History:** PLOS recognizes the
benefits of transparency in the peer review
process; therefore, we enable the publication
of all of the content of peer review and
author responses alongside final, published
articles. The editorial history of this article is
available here: https://doi.org/10.1371/journal.
pntd.0014160

## Abstract

### Background

Primaquine (PQ) is the only widely available drug that eliminates latent hypnozoites,
thereby preventing relapses of *Plasmodium vivax* malaria. Because PQ biotransfor-
mation mediated by the cytochrome P-450 (CYP) isoenzyme CYP2D6 is required for
therapeutic activity, patients with reduced CYP2D6 activity can experience relapse
despite treatment. The minimum level of CYP2D6 activity for adequate anti-relapse
PQ efficacy is unknown.

### Methods and findings

We conducted a retrospective, population-based cohort study in the main malaria
hotspot of Brazil. We fitted time-to-event data from cohort participants who experi-
enced vivax malaria, using Cox proportional hazards models, to explore how
genotype-determined CYP2D6 activity, expressed as activity scores (AS), modulates
the risk of *P. vivax* recurrence within 6 months after treatment with chloroquine and
PQ (total dose, 3.5 mg/kg). We analyzed community-wide *P. vivax* malaria incidence
data, using a multivariable negative binomial regression model, to quantify the

**Data availability statement:** All relevant data are within the paper and its Supporting information files.

**Funding:** This work was supported by grants from FAPESP (2016/18740-9, 2022/11963-3, and 2024/04280-2 to MUF), the National Institute of Allergy and Infectious Diseases of the National Institutes of Health, United States (U19 AI089681 to MUF), and the Fundação para a Ciência e Tecnologia (FCT), Portugal, through the institutional projects UID/04413/2020 and LA-REAL LA/P/0117/2020 (MUF), and the Fundação de Amparo à Pesquisa do Estado de Minas Gerais, Brazil (APQ-00516-21 to TNS). MUF and TNS are supported by senior researcher scholarships from the Conselho Nacional de Desenvolvimento Científico e Tecnológico of Brazil (CNPq), Brazil. RMS is supported by the Eric and Wendy Schmidt AI in Science Global Faculty fellowship. MCSBP is supported by scholarships from the Coordenação de Aperfeiçoamento de Pessoal de Nível Superior (CAPES) of Brazil (Finance Code 001) and the René Rachou Institute of Fiocruz, Brazil. CAPJ (2024/10682-6), WAL (2023/15369-1), and PTR (2018/03902-9) are/were supported by scholarships from the Fundação de Amparo à Pesquisa do Estado de São Paulo (FAPESP), Brazil. The funders had no role in study design, data collection and analysis, decision to publish, or preparation of the manuscript.

**Competing interests:** The authors have declared that no competing interests exist.

impact of reduced CYP2D6 activity on the overall risk of vivax malaria, whether from relapses or new infections. Among 466 *P. vivax*-infected cohort participants, those with null/low CYP2D6 activity (AS ≤ 0.25), but not participants with intermediate CYP2D6 activity (AS from 0.5 to 1.0), had twice the risk of recurrence compared to an AS > 1.0 (hazard ratio = 2.22, $P = 0.004$). However, vivax malaria incidence did not differ significantly across CYP2D6 activity levels during 5 years of follow-up of 997 Amazonians exposed to intense transmission (mean, 26.6 vivax malaria cases/100 person-years). These findings suggest that the excess of relapses among people with severely reduced CYP2D6 activity adds relatively little to the overall burden of vivax malaria episodes.

## Conclusion

Amazonians with an AS ≤ 0.25, but not necessarily those with intermediate CYP2D6 activity, have a greater risk of recurrence after a PQ-treated *P. vivax* infection and require alternative relapse suppression regimens for the radical cure of vivax malaria.

## Author summary

The cytochrome P-450 (CYP) isoenzyme 2D6 metabolizes up to 25% of commonly used medicines, including primaquine (PQ), which is the only widely available drug that eliminates latent hypnozoites, thus preventing relapses of *Plasmodium vivax* malaria. Biotransformation mediated by CYP2D6 is required for the therapeutic activity of PQ and patients with reduced CYP2D6 activity can experience relapsing *P. vivax* infections despite treatment. However, the minimum level of CYP2D6 activity for adequate anti-relapse efficacy of PQ remains unknown. Here, we show that vivax malaria patients with null or very low CYP2D6 activity – genotype-determined activity score (AS) ≤ 0.25 – treated with chloroquine and low-dose PQ (total dose, 3.5 mg/kg over 7 or 14 days) are at a significantly greater risk of recurrence within six months after treatment, compared to those with normal/high CYP2D6 activity (AS > 1.0). Surprisingly, however, the overall incidence of *P. vivax* infection over 5 years did not differ significantly according to CYP2D6 activity. The radical cure of vivax malaria patients with an AS ≤ 0.25 requires relapse suppression regimens alternative to low-dose PQ, such as high-dose PQ, tafenoquine, or weekly chloroquine prophylaxis.

## Introduction

The relatively neglected human parasite *Plasmodium vivax* accounts for approximately 12.4 million annual malaria cases globally [1] and nearly 90% of the malaria burden in Brazil [2]. *P. vivax* can be particularly challenging to control and eliminate due to its unique biology [3]. One of its distinctive features is the ability to stay

dormant in the liver as hypnozoites that may give rise to one or more malaria episodes known as relapses, often several weeks or months following the primary infection.

The widely available 8-aminoquinoline primaquine (PQ) or the recently introduced tafenoquine should be administered along with blood schizonticidal therapy, such as chloroquine (CQ) or artemisinin-based combination therapies, to clear hepatic hypnozoites and achieve the radical cure of vivax malaria [4]. Importantly, PQ is a prodrug that needs to be activated by the cytochrome P-450 (CYP) isoenzyme 2D6 (CYP2D6) for therapeutic activity; more specifically, CYP2D6-mediated hydroxylation generates metabolites with redox activity that account for the antimalarial activity of PQ [5]. It has recently been suggested that the massive, newly discovered, hidden extrahepatic reservoir of asexual *P. vivax* stages in the spleen and bone marrow might also be targeted by PQ [6]. There are more than 160 known allelic variants of the *CYP2D6* gene that are associated with metabolic activities ranging from poor to ultrarapid [7]. As a consequence, naturally occurring polymorphism in the CYP2D6 isoenzyme can prevent or reduce PQ activation and lead to relapses despite PQ administration, as first described in naïve adults treated with supervised CQ and high-dose PQ following a challenge with *P. vivax* sporozoites [8]. Whether the therapeutic activity of tafenoquine also requires CYP2D6-dependent activation is unclear (S1 File) [9].

From a clinical perspective, identifying the CYP2D6 activity threshold associated with an increased risk of PQ failure is critical to inform the use of alternative anti-relapse regimens. However, the estimated impact of low CYP2D6 activity on PQ efficacy ranges from none to substantial among studies to date (S1 Table and S1 File). For example, three clinical trials from South America failed to show a significant difference in the frequency of *P. vivax* malaria recurrences following CQ-PQ treatment in patients with a predicted CYP2D6 activity score (AS) ≤ 1.0, compared to patients with an AS greater than 1.0 [10–12]. By contrast, an AS ≤ 1.0 was associated with more frequent *P. vivax* malaria recurrences in some [13–15], although not all observational studies from this region [16]. A likely source of variability across studies is acquired immunity [14]. Carriers of low-activity CYP2D6 variants are hypothesized to develop immunity to blood-stage *P. vivax* infection faster than those with normal CYP2D6 activity, due to repeated relapses since birth, becoming semi-immune as adults [17].

Here, we examine how reduced CYP2D6 activity affects *P. vivax* infection risk in a large cohort of Amazonians exposed to intense malaria transmission. We show that patients with severely reduced CYP2D6 activity (AS ≤ 0.25), once infected with *P. vivax,* have a greater risk of symptomatic recurrence within six months after treatment. However, the overall incidence of vivax malaria over 5 years of follow-up does not differ according to CYP2D6 activity levels.

## Methods

### Ethics statement

Study protocols were approved by the Institutional Review Boards of the Institute of Biomedical Sciences, University of São Paulo (CAAE 64767416.6.0000.5467), and the René Rachou Research Center of Fiocruz (2.803.756), Belo Horizonte, both in Brazil. Written informed consent and assent were obtained from all participants in the Mâncio Lima Cohort Study or their parents/guardians.

### Study area and population

The town of Mâncio Lima (7°36′28.6" S, 72°54′23.0" W) is the main urban malaria hotspot of Brazil, with an annual parasite incidence of 320.7 cases per 1,000 inhabitants in 2018, the highest of any urban center in this country (S1 Fig.; S1 File). Between November 2015 and April 2016, we enumerated 9,124 residents in Mâncio Lima distributed into 2,329 households [18]. Starting in April 2018, we invited residents in 20% of the local households to participate in an open cohort study on biological and sociodemographic determinants of malaria risk [18]. Of 2,774 participants enrolled over the following 4 years [19], DNA samples were available for 1,862 and 1,273 had their *CYP2D6* genotypes inferred [17].

## Retrospective cohort

S2 Fig summarizes the steps for retrospective cohort construction. We included study participants with known Duffy blood group (*FY*) and *CYP2D6* genotypes who resided in Mâncio Lima anytime between 2014 and 2018. We excluded Duffy-negative participants carrying the T-67C nucleotide substitution in the globin transcription factor–1 (GATA–1) binding motif of the *ACKR1* gene (S1 File), which suppresses Duffy blood group antigen expression on the red blood cell surface. This is because they are partially or completely resistant to *P. vivax* blood-stage infection [20], which might confound subsequent analyses. We tested 1,921 study participants for the *ACKR1* genotype. Of those, one participant was next excluded due to missing age information, 183 were excluded because they did not reside in the study area anytime between January 2014 and December 2018, 680 because we lacked complete *CYP2D6* genotype information, and 106 because they were Duffy-negative (i.e., homozygous for the T-67C GATA–1 mutation). After all exclusions, 997 study participants were retained in the retrospective cohort (S2 Fig). The date of entry in the retrospective cohort was the date of birth, the date the participant moved to Mâncio Lima, for those not counted in the 2015–16 census, or January 1, 2014 – whichever was the most recent. We used the dates of entry and exit to estimate the number of person-years at risk; participants who left the study area before December 31, 2018, were considered lost to follow-up since the date they moved away.

The study outcome was any single-species *P. vivax* infection, irrespective of parasite density and the presence of symptoms, diagnosed by thick-smear microscopy or (rarely) by rapid diagnostic test from January 1, 2014, through December 31, 2018. Because the vast majority of infections were symptomatic (see Results), we can alternatively define our study outcome as vivax malaria – i.e., symptomatic infection with *P. vivax*. We searched the electronic malaria notification database of the Ministry of Health of Brazil [21] for cases recorded during the study period that matched cohort participants' name, sex, and age, and their mothers' name (S2 Fig and S1 File).

An interval equal to or greater than 28 days between two or more consecutive episodes was required to count the latter episode as a new infection. When different plasmodial species were identified in separate samples collected less than 28 days apart, the participant was considered to have had a single episode of mixed-species infection. *P. vivax* infections were routinely treated with CQ (25 mg/kg over 3 days) and PQ (3.5 mg/kg over 7 days), except for cases of PQ ineligibility such as those of glucose-6-phosphate dehydrogenase deficiency, pregnancy, lactation, or age < 6 months (S1 File).

## *CYP2D6* genotyping

We used custom OpenArray assays (Thermo Fisher Scientific, Waltham, MA) to identify nine single-nucleotide polymorphisms and three deletions at the *CYP2D6* locus. Metabolic activity values were assigned to individual *CYP2D6* alleles according to the Pharmacogene Variation Consortium (PharmVar) guidelines (https://www.pharmvar.org/gene/CYP2D6); an activity value of 1 corresponds to fully functional alleles, while 0 corresponds to non-functional alleles, and values between 0.25 and 0.5 to reduced-activity alleles. The Hs00010001_cn assay (Thermo Fisher Scientific) was used esti-mate the *CYP2D6* gene copy number; activity values were multiplied when multiple *CYP2D6* allele copies were present (S1 File). We used the Haplo2D6 web tool to automate haplotype assignment, activity score calculation, and phenotype inference (https://bioinfo.dcc.ufmg.br/Haplo2D6/). Activity values assigned to individual *CYP2D6* alleles were summed to obtain a genotype-determined AS [7].

## Statistical analysis

Statistical analysis was carried out in R 4.2.4 (R Foundation for Statistical Computing, Vienna, Austria) and statistical significance was set at the 5% level.

First, we ran a time-to-recurrence analysis nested within the retrospective cohort study, using the R package *survival*. We identified the first or only vivax malaria episode experienced by participants between 2014 and 2018 and excluded

individuals not treated with PQ or given a blood schizonticidal drug partner other than CQ (S3 Fig and S1 File). The end-point was the first or only *P. vivax* malaria recurrence diagnosed between days 28 and 180 after starting CQ-PQ treatment, which may be due to either a relapse from dormant hypnozoites (PQ treatment failure) or a new infection following *P. vivax* sporozoite inoculation. Late recrudescence due to schizonticide failure seems unlikely given the very high CQ efficacy in the study area [10,22]. Survival times of patients who did not experience a *P. vivax* malaria recurrence were right-censored at the end of the 180-day follow-up period, at the time the patient left the study area, or on December 31, 2018, whichever came first. Likewise, survival times of patients who experienced a non-vivax malaria episode prior to any *P. vivax* malaria recurrence were right-censored at the time of diagnosis. A log-rank test was used to compare survival curves.

Cox proportional hazards models were used to estimate hazard ratios (HRs), along with 95% confidence intervals (CIs), for the association between CYP2D6 activity and risk of *P. vivax* recurrence, while adjusting for sex, age at the time of diagnosis (0–16, 17–40, and >40 years), *FY* genotype (*FY*01/FY*01, FY*01/FY*01N.01, FY*02/FY*02, FY*02/FY*01N.01,* or *FY*01/FY*02*) according to the International Society of Blood Transfusion (https://www.isbtweb.org/resource/008fy.html), and socioeconomic status using wealth index terciles as a proxy [19]. The proportional hazards assumption of the Cox model was evaluated using Schoenfeld residuals.

Next, we examined the association between CYP2D6 activity and *P. vivax* malaria incidence rates. The outcome was any *P. vivax* infection diagnosed in cohort participants between 2014 and 2018. We used the R package *gamlss* to fit incidence data with a multivariable negative binomial regression model [23]. We estimated incidence rate ratios (IRR), along with 95% confidence intervals, to quantify the impact of CYP2D6 activity (AS ≤ 0.25 *vs* AS ≥ 0.50) on vivax malaria incidence, while adjusting for sex, age on December 31, 2018 (stratified as above), *FY* genotype, and wealth index terciles. Because malaria incidence varied over time and participants differed as regards follow-up duration, a covariate that represents an individual's time at risk weighted by daily malaria incidence was added.

### Role of the funding source

The funders of the study had no role in study design, data collection, data analysis, data interpretation, or writing of the report.

### Results

The retrospective cohort comprised 997 people aged between <1 and 101 years (mean, 29.1 years), who contributed 4497.6 person-years of follow-up (Fig 1). The study population did not differ significantly, according to age, sex, *FY* genotype, and wealth index, from the 634 Fy-positive participants in the original Mâncio Lima cohort who were not included in the present analysis due to missing *CYP2D6* genotype information (S3 Table). The following *CYP2D6* alleles associated with low or null activity were detected: *3, *4, *5, *6, *9, *10, *17, *29,* and *41* [17]. Only 45 (4.5%; 95% CI, 3.4–6.0%) study participants had a predicted CYP2D6 activity of zero and 279 (28.0%; 95% CI, 25.3–30.9%) had an AS ≤ 1.0, the threshold commonly used to distinguish bewteen poor/intermediate and normal/ultrarapid metabolizers [7] (Table 1). We retrieved 1,197 *P. vivax* infections in the study population between January 2014 and December 2018; 98.5% were diagnosed through passive case detection in symptomatic patients. The average incidence of vivax malaria was 26.6/100 (95% CI, 25.1–28.2/100) person-years at risk, with an overdispersed distribution of infections per participant – mean of 1.2 (range of 0–11) and variance of 2.9. While 516 (51.7%) participants remained free of vivax malaria over 5 years, 185 (18.6%) had three or more infections each and together contributed 782 vivax malaria cases, 65.3% of the total (S4 Fig).

### CYP2D6 activity and time to *Plasmodium vivax* recurrence

The time-to-event analysis comprised 466 people aged 1–101 years (mean, 29.5 years) who had at least one *P. vivax* infection diagnosed between January 2014 and December 2018; 131 (28.1%) had one or more recurrences within 6

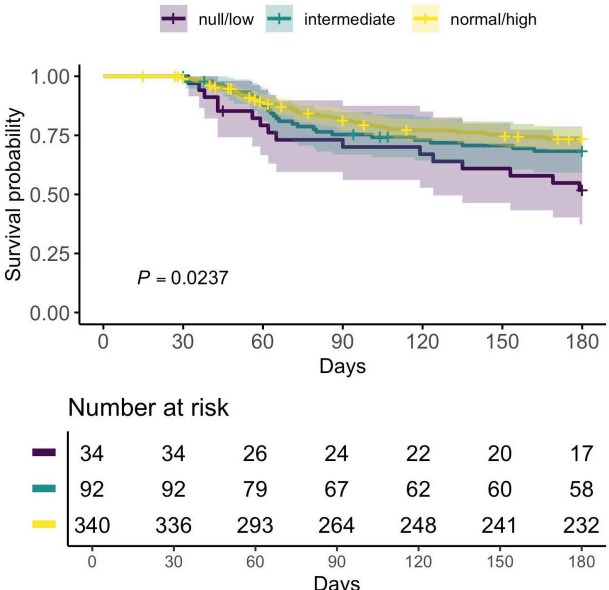

**Fig 1. Kaplan-Meier survival curves for 466 study participants with at least one vivax malaria episode treated with chloroquine-primaquine between 2014 and 2018.** Three CYP2D6 activity levels were defined: null/low (genotype-determined activity score [AS] ≤ 0.25), intermediate (AS between 0.5 and 1.0), and normal/high (AS > 1.0). The shaded areas surrounding the lines represent 95% confidence intervals. The *P* value of 0.0237 was obtained with a log-rank test comparing the three survival curves, rejecting the null hypothesis that groups with different CYP2D6 activity levels share an identical survival curve.

**Table 1. Distribution of cytochrome P-450 (CYP) 2D6 activity scores predicted from genotypes among 997 Mâncio Lima cohort participants, 2014–18.**

| Activity Score | CPIC Classification | No. of participants | Frequency, % (95% CI) | Cumulative frequency, % (95% CI) |
|---|---|---|---|---|
| 0.00 | Poor metabolizer | 45 | 4.5 (3.4, 6.0) | 4.5 (3.4, 6.0) |
| 0.25 | Intermediate metabolizer | 25 | 2.5 (1.7, 3.7) | 7.0 (5.6, 8.8) |
| 0.50 | Intermediate metabolizer | 15 | 1.5 (0.9, 2.5) | 8.5 (6.9, 10.4) |
| 0.75 | Intermediate metabolizer | 7 | 0.7 (0.3, 1.4) | 9.2 (7.6, 11.2) |
| 1.0 | Intermediate metabolizer | 187 | 18.8 (16.5, 21.3) | 28.0 (25.3, 30.9) |
| 1.25 | Normal metabolizer | 126 | 12.6 (10.7, 14.8) | 40.6 (37.6, 43.7) |
| 1.50 | Normal metabolizer | 36 | 3.6 (2.6, 5.0) | 44.2 (41.2, 47.3) |
| 2.00 | Normal metabolizer | 460 | 46.1 (43.1, 49.2) | 90.4 (88.4, 92.1) |
| >2.25 | Ultrarapid metabolizer | 96 | 9.6 (7.9, 11.6) | 100.0 (99.6, 100.0) |

CPIC = Clinical Pharmacogenetics Implementation Consortium [7]; CI = confidence interval.

months of CQ-PQ treatment. Since the usual threshold of AS ≤ 1.0 may not predict PQ failure in Amazonians (S1 Table and S1 File), we estimated adjusted hazard ratios for the association between time to first recurrence and CYP2D6 activity using six different AS thresholds, ranging from 0 to 1.25 (S2 Table). We found that the differences in recurrence risk were gradually attenuated by raising the AS threshold from 0.25 to 1.0, and lost statistical significance at an AS = 1.25 (Table 2; S4 Table).

We next compared times to recurrence across three CYP2D6 activity levels: null/low (AS ≤ 0.25), intermediate (AS between 0.5 and 1.0), and normal/high (AS > 1.0). Kaplan-Meier survival curves are shown in Fig 1. The proportions of

**Table 2. Association between low cytochrome P-450 (CYP) 2D6 activity, defined with six different activity score (AS) thresholds, and the time to the first or only *Plasmodium vivax* malaria recurrence, as estimated by Cox proportional hazards models, among 466 Mâncio Lima cohort participants, 2014–18.**

| AS threshold | No. of participants in the "low-activity" group | No. of participants in the "normal-activity" group | Hazard ratio (95% confidence interval) | *P* value |
|---|---|---|---|---|
| 0.00 | 45 | 421 | 2.01 (1.01, 3.98) | 0.046 |
| ≤ 0.25 | 70 | 396 | 2.10 (1.23, 3.57) | 0.006 |
| ≤ 0.50 | 85 | 381 | 1.91 (1.14, 3.22) | 0.015 |
| ≤ 0.75 | 92 | 374 | 1.69 (1.01, 2.83) | 0.047 |
| ≤ 1.00 | 279 | 187 | 1.52 (1.05, 2.21) | 0.027 |
| ≤ 1.25 | 405 | 61 | 1.06 (0.74, 1.51) | 0.752 |

participants who remained recurrence-free at day 180 were 51.8% (95% CI, 37.3 to 72.0%) in the null/low activity group, 68.3% (95% CI, 59.3 to 78.8%) in the intermediate activity group, and 73.5% (95% CI, 68.8 to 78.4) in the normal/high activity group ($P = 0.0237$). A Cox regression model adjusted for sex, age at the time of diagnosis, *FY* genotype, and wealth index tercile showed no statistically significant difference between participants with intermediate vs. normal/high CYP2D6 activity, while the time to recurrence was significantly shortened among participants with null/low activity (hazard ratio = 2.22, $P = 0.004$; Table 3). These results suggest that an AS between 0.5 and 1.0 may suffice for PQ activation and efficacy against symptomatic relapse, while participants with an AS ≤ 0.25 are at significantly increased risk of symptomatic recurrence following infection with *P. vivax*.

### CYP2D6 activity and *Plasmodium vivax* infection incidence

We next examined the impact of null/low CYP2D6 activity on vivax malaria incidence over 5 years. Somewhat surprisingly, we observed similar incidence rates of symptomatic *P. vivax* infection from 2014 to 2018 among study participants with null/low, intermediate, and normal/high CYP2D6 activity, ranging between 25.6 and 28.9 per 100 person-years of follow-up (Table 4). Vivax malaria incidence varied markedly across age groups. An unadjusted, age-stratified analysis suggested that children and adolescents with null/low CYP2D6 activity were at increased risk of infection, compared to those with intermediate and normal/high CYP2D6 activity (Table 4). No such difference was apparent among older participants.

We next tested whether increasing age, a proxy for cumulative exposure to malaria, might attenuate the effect of reduced CYP2D6 activity on vivax malaria risk. To this end, we fitted a multivariable negative binomial regression model to *P. vivax* infection incidence data, while adjusting for sex, age, *FY* genotype, wealth index tercile, and log-transformed individual's time at risk weighted by daily malaria incidence, and tested for an interaction between age and CYP2D6 activity. The association between null/low CYP2D6 activity (AS ≤ 0.25) and vivax malaria incidence did not reach statistical significance among the youngest participants (IRR = 1.65, $P = 0.080$) and did not change significantly in the older age groups ($P = 0.090$ for 17–40 years and $P = 0.343$ for > 40 years) (Table 5). Therefore, we found no statistically significant effect of age on the association between CYP2D6 activity and vivax malaria incidence in our study population.

### Discussion

This is the largest longitudinal study to date on the association between CYP2D6 activity and the risk of vivax malaria recurrence. Our time-to-event analysis shows that vivax malaria patients with severely reduced CYP2D6 activity (AS ≤ 0.25; 7.0% of study participants) have twice the risk of symptomatic recurrence within six months after CQ-PQ treatment, compared to those with an AS > 1.0. We expected that differences in CYP2D6 activity would contribute significantly to the marked individual variation in vivax malaria incidence seen in the Amazon, where 20% of high-risk people experience more than 80% of the infections [24]. However, contrary to our expectation, the overall incidence of symptomatic *P. vivax*

**Table 3. Association between cytochrome P-450 (CYP) 2D6 activity and time to the first or only symptomatic *P. vivax* malaria recurrence within 6 months of CQ-PQ treatment, as estimated by a Cox proportional hazards model, among 466 Mâncio Lima cohort participants, 2014–18.**

| Variable | No. of participants | Adjusted hazard ratio | 95% confidence interval | *P* value |
|---|---|---|---|---|
| Age | | | | |
| 0–16 years | 124 | 1.00 | Reference | |
| 17–40 years | 229 | 0.91 | 0.62, 1.33 | 0.621 |
| >40 years | 113 | 0.62 | 0.36, 1.06 | 0.078 |
| Sex | | | | |
| Female | 243 | 1.00 | Reference | |
| Male | 223 | 1.34 | 0.94, 1.90 | 0.101 |
| Wealth index tercile | | | | |
| 1 (poorest) | 170 | 1.00 | Reference | |
| 2 | 165 | 1.40 | 0.93, 2.13 | 0.109 |
| 3 (wealthiest) | 131 | 1.38 | 0.87, 2.18 | 0.169 |
| *FY* genotype | | | | |
| *FY*01/FY*01N.01* | 80 | 1.00 | Reference | |
| *FY*01/FY*01* | 67 | 0.69 | 0.38, 1.25 | 0.224 |
| *FY*02/ FY*01N.01* | 71 | 0.62 | 0.33, 1.14 | 0.124 |
| *FY*02/FY*02* | 82 | 0.81 | 0.46, 1.41 | 0.454 |
| *FY*01/FY*02* | 166 | 0.64 | 0.39, 1.06 | 0.084 |
| CYP2D6 activity score range | | | | |
| 0.00 to 0.25 | 34 | 2.22 | 1.29, 3.83 | 0.004 |
| 0.50 to 1.00 | 92 | 1.29 | 0.83, 1.99 | 0.258 |
| >1.00 | 340 | 1.00 | Reference | |

infection diagnosed over 5 years did not differ according to CYP2D6 activity levels, suggesting that relapses add relatively little to the overall burden of vivax malaria experienced by cohort participants with reduced CYP2D6 activity.

We have recently observed similar prevalence rates of *P. vivax* infection across CYP2D6 activity levels in the general population of Mâncio Lima [17]. Nevertheless, an age-stratified analysis revealed a greater *P. vivax* infection prevalence among adolescents and young adults with normal CYP2D6 activity, compared to those with an AS < 1.0 [17]. We hypothesized that frequent relapses during the first decade of life would accelerate the development of immunity to blood-stage parasites and render carriers of low-activity CYP2D6 variants more likely to suppress relapsing parasitemia as adolescents and young adults, compared to normal and ultrarapid metabolizers who experienced less frequent relapses since birth, due to the routine use of PQ [17]. In other words, age – a proxy of immunity associated with cumulative exposure to malaria parasites – would act as an effect modifier in the association between CYP2D6 activity and *P. vivax* infection risk [25]. Here, we failed to corroborate the hypothesis of effect modification, by analyzing the incidence of predominantly symptomatic *P. vivax* infections diagnosed by microscopy or rapid diagnostic tests during high-transmission years, from 2014 to 2018 (Table 5). By contrast, the prevalence study [17] considered a different outcome: predominantly asymptomatic and submicroscopic *P. vivax* infections diagnosed with a highly sensitive PCR during consecutive cross-sectional surveys carried out at times of decreasing transmission, between 2018 and 2021 [19]. Whether the effect of reduced CYP2D6 activity on community-wide vivax malaria risk is modified by age remains unresolved (S1 File).

Reduced CYP2D6 activity has been associated with PQ treatment failure since 2013 [8], but the minimum levels of CYP2D6 activity needed for PQ activation and anti-relapse efficacy across different settings remain unknown [8,25]. Recurrence rates of *P. vivax* malaria have been compared between normal/ultrarapid and poor/intermediate metabolizers (S1 Table), but the heterogeneous group of patients with an AS between 0 and 1.0 displays widely different phenotypes,

**Table 4. Incidence of *P. vivax* malaria per 100 person-years of follow-up, according to cytochrome P-450 (CYP) 2D6 activity and age, among 997 Mâncio Lima cohort participants, 2014–18.**

| CYP2D6 activity score range | No. of participants | Person-years | *P. vivax* infection incidence (95% CI) |
|---|---|---|---|
| **Age 0–16 years** | | | |
| 0.00 to 0.25 | 29 | 117.59 | 34.87 (25.02, 47.30) |
| 0.50 to 1.00 | 61 | 245.32 | 18.34 (13.38, 24.55) |
| >1.00 | 233 | 994.56 | 23.13 (20.23, 26.32) |
| All CYP2D6 activity strata | 323 | 1357.47 | 23.28 (20.78, 25.99) |
| **Age 17–40 years** | | | |
| 0.00 to 0.25 | 25 | 110.42 | 27.17 (18.33, 38.78) |
| 0.50 to 1.00 | 79 | 367.65 | 31.82 (26.32, 38.14) |
| >1.00 | 313 | 1444.41 | 31.71 (28.87, 34.75) |
| All CYP2D6 activity strata | 417 | 1922.49 | 31.47 (29.01, 34.08) |
| **Age >40 years** | | | |
| 0.00 to 0.25 | 16 | 80.00 | 22.50 (13.33, 35.56) |
| 0.50 to 1.00 | 69 | 331.81 | 24.11 (19.12, 30.01) |
| >1.00 | 172 | 805.78 | 22.09 (18.96, 25.58) |
| All CYP2D6 activity strata | 257 | 1217.60 | 22.67 (20.07, 25.51) |
| **All age groups** | | | |
| 0.00 to 0.25 | 70 | 308.01 | 28.89 (23.20, 35.56) |
| 0.50 to 1.00 | 209 | 944.78 | 25.61 (22.49, 29.05) |
| >1.00 | 718 | 3244.76 | 26.69 (24.94, 28.53) |
| All CYP2D6 activity strata | 997 | 4497.56 | 26.61 (25.13, 28.17) |

Note: CI = confidence interval.

from null to near-normal CYP2D6 activity [7]. Here, we found no statistically significant difference in the time to recurrence between intermediate and normal or ultrarapid metabolizers, indicating that only patients with more extreme phenotypes (AS ≤ 0.25) are at a significantly increased risk of recurrence due to PQ failure (Table 3).

Our findings may not hold for all human populations exposed to *P. vivax*. First, very few patients with an AS ≤ 0.25 have been included to date in worldwide time-to-recurrence studies to determine the critical levels of CYP2D6 activity that cause PQ failure in different populations [9,26]. Second, the well-known regional variation in *P. vivax* response to PQ [4,27] might contribute to vivax malaria recurrences among patients with near-normal CYP2D6 activity treated with low-dose PQ in certain endemic settings [28]. Third, human populations differ widely in the intensity of exposure to *P. vivax* and levels of acquired immunity, which may modulate relapse risk [14]. Importantly, vivax malaria patients with CYP2D6 activity low enough to reduce the efficacy of low-dose PQ should be identified and prescribed alternative relapse suppression regimens, such as high-dose PQ [29], tafenoquine [30], or weekly chloroquine prophylaxis [31].

This study has four main limitations. First, malaria case records were retrieved retrospectively from a (mostly) passive surveillance database and no blood samples were available for further confirmatory diagnostic testing. Our analysis is essentially limited to symptomatic *P. vivax* infections that prompted a visit to a health facility and were diagnosed by microscopy or rapid diagnostic test; as a consequence, we overlook submicroscopic and most asymptomatic infections. Second, passive surveillance is prone to biases due to differences in access to health facilities and individual health-seeking behavior. Third, our time-to-event analysis does not distinguish relapses due to PQ failure from new infections and (less likely) late recrudescences, collectively defined as recurrences. Relapse frequencies were not specifically measured here, but modeling approaches of varying complexity are now available to distinguish relapses from new infections

**Table 5. Factors associated with vivax malaria incidence density, as identified by multivariable negative binomial regression analysis, among 997 Mâncio Lima cohort participants, 2014–18.**

| Variable | No. of participants | Adjusted incidence rate ratio | 95% confidence interval | P value |
|---|---|---|---|---|
| Age | | | | |
| 0–16 years | 323 | 1.00 | Reference | |
| 17–40 years | 417 | 1.51 | 1.20, 1.91 | <0.001 |
| >40 years | 257 | 1.11 | 0.85, 1.45 | 0.434 |
| Sex | | | | |
| Female | 525 | 1.00 | Reference | |
| Male | 472 | 1.08 | 0.90, 1.29 | 0.424 |
| Wealth index tercile | | | | |
| 1 (poorest) | 347 | 1.00 | Reference | |
| 2 | 348 | 0.83 | 0.67, 1.02 | 0.089 |
| 3 (wealthiest) | 302 | 0.67 | 0.53, 0.85 | <0.001 |
| *FY* genotype | | | | |
| *FY*01/FY*01N.01* | 173 | 1.00 | Reference | |
| *FY*01/FY*01* | 155 | 1.01 | 0.72, 1.42 | 0.944 |
| *FY*02/ FY*01N.01* | 172 | 0.95 | 0.70, 1.29 | 0.756 |
| *FY*02/FY*02* | 170 | 1.11 | 0.81, 1.52 | 0.498 |
| *FY*01/FY*02* | 327 | 1.15 | 0.87, 1.52 | 0.311 |
| CYP2D6 activity score range (for age 0–16 years) | | | | |
| 0.00 to 0.25 | 29 | 1.65 | 0.93, 2.92 | 0.080 |
| >0.25 | 294 | 1.00 | – | |
| Interaction age: CYP2D6 activity score | | | | |
| Age 17–40: 0.00 to 0.25 | 25 | 0.50 | 0.22, 1.14 | 0.090 |
| Age 17–40:>0.25 | 392 | – | – | |
| Age>40: 0.00 to 0.25 | 16 | 0.64 | 0.26, 1.58 | 0.343 |
| Age>40: >0.25 | 241 | – | – | |
| Time at risk, weighted by incidence (log) | | 2.78 | 2.02, 3.83 | <0.001 |

[32,33]. Moreover, treatment was not directly observed and adherence to the relatively lengthy CQ-PQ regimen was not monitored. Fourth, we assume that cohort participants gradually develop clinically immunity as they become more exposed to malaria with increasing age, but did not measure laboratory correlates of acquired immunity (e.g., levels of IgG antibodies to key *P. vivax* antigens) to test this hypothesis in the study population.

In summary, we show that patients with an AS ≤ 0.25, but not necessarily those with intermediate CYP2D6 activity, have a significantly increased risk of vivax malaria recurrence within 6 months after the treatment of a *P. vivax* infection with CQ and low-dose PQ, compared to patients with an AS > 1.0. This has practical implications for the radical cure of vivax malaria.

## Supporting information

**S1 Checklist. STROBE checklist (doi: 10.1097/EDE.0b013e3181577654).**
(DOCX)

**S1 Fig. Study area.** *A,* Location of the municipality of Mâncio Lima (black) in Acre State (gray) in the western part of Brazil (light gray), next to the border with Peru. *B,* Aerial photography of the municipality seat, the town of Mâncio Lima, taken by the first author. The map in panel A was created with QGIS software version 3.14, an open-source Geographic

Information System (GIS) licensed under the GNU General Public License (https://bit.ly/2BSPB2F). Publicly available shape files provided from the Brazilian Institute of Geography and Statistics (IBGE) website (https://bit.ly/34gMq0S). All geographical data are used under the Creative Commons Attribution License (CC BY 4.0).
(PDF)

**S2 Fig. Study flow diagram.** Finger-prick blood samples were collected from 2,774 participants in the Mâncio Lima cohort study between April 2018 and November 2021. The Duffy blood group (*FY*) genotype was determined for 1,921 participants. Of those, one participant was excluded due to missing age information, 183 were excluded because they did not reside in the study area anytime between January 2014 and December 2018, 680 because complete *CYP2D6* genotype information was not available, and 106 because they were Duffy (Fy)-negative. After all exclusions, 997 study participants remained in the retrospective cohort.
(PDF)

**S3 Fig. Main study outcome.** Malaria case records from 01 January 2014 through 31 December 2018 were retrieved from the SIVEP-Malaria database and matched to study participants.
(PDF)

**S4 Fig. Participants in the time-to-event analysis.** We identified the first or only symptomatic *P. vivax* infection experienced by 481 participants between 2014 and 2018 and excluded 15 episodes that were not treated with PQ or given a blood schizonticidal partner drug other than CQ. There were 466 participants in the time-to-recurrence analysis and 131 first or only *P. vivax* malaria recurrences were diagnosed during the 6-month follow-up period.
(PDF)

**S5 Fig. Distribution of the number of *Plasmodium vivax* infections per study participant between 1 January 2014 and 31 December 2018.** Case records were retrieved from the SIVEP-Malaria database and matched to study participants (n = 997).
(PDF)

**S1 Table. Summary of published studies on CYP2D6 activity and risk of *Plasmodium vivax* recurrence.**
(PDF)

**S2 Table. Classical CYP2D6 phenotype classification and alternative classifications used in the present study.**
(PDF)

**S3 Table. Characteristics of Duffy (Fy)-positive Mâncio Lima cohort participants included in the present analysis (*n* = 997) and those excluded from the present analysis due to missing *CYP2D6* genotype information (*n* = 634).**
(PDF)

**S4 Table. Distribution of cytochrome P-450 (CYP) 2D6 activity scores predicted from genotypes among 997 Mâncio Lima cohort participants, 2014–18.**
(PDF)

**S5 Table. Association between different activity score thresholds to define "low" CYP2D6 activity and the risk of *P. vivax* malaria recurrence within 6 months of CQ-PQ treatment, as estimated by logistic regression analysis, among 466 Mâncio Lima cohort participants, 2014–18.**
(PDF)

**S1 File. Supplementary Methods, Results, and Discussion.**
(DOCX)

**S1 Data. CSV file with all variables used in retrospective cohort study analysis.**
(CSV)

**S2 Data. CSV file with all variables used in the time-to-recurrence analysis.**
(CSV)

## Acknowledgments

We are grateful to the participants in the Mâncio Lima Cohort Study for their enthusiastic support. We acknowledge the logistic support provided by Ajucilene (Joice) G. Mota, Francisco Melo, and their team at the Health Secretariat of Mâncio Lima. We thank the Program for Technological Development in Tools for Health-PDTIS FIOCRUZ for the use of the Real-Time PCR Facility (RPT09D) at the René Rachou Institute of Fiocruz.

**Mâncio Lima Cohort Study Working Group:** Alexandre S. Nogueira, Anderson R. J. Fernandes, Andreea-Beatrice Rusu, Bárbara Prado C. Silva, Igor C. Johansen, Isabel Giacomini, Jaques N. de Carvalho, Juliana C. Belizário, Juliana Tonini, Lais C. Salla, Marcelo U. Ferreira, Maria José Menezes, Pablo S. Fontoura, Priscila R. Calil, Priscila T. Rodrigues, Rodrigo M. Corder, Thaís C. de Oliveira, Vanessa C. Nicolete, and Winni A. Ladeia (University of São Paulo, São Paulo, Brazil); Amanda O. S. Fernandes and Rodrigo M. Martorano (Federal University of Acre, Cruzeiro do Sul, Brazil); Paulo E. M. Ribolla (State University of São Paulo, Botucatu, Brazil); Simone Ladeia-Andrade (Oswaldo Cruz Institute, Fiocruz, Rio de Janeiro, Brazil); Carlos E. Cavasini (Faculdade de Medicina de São José do Rio Preto, São José do Rio Preto, Brazil); Joseph M. Vinetz (Yale School of Medicine, New Haven, Connecticut, USA); and Marcia C. Castro (Harvard T. H. Chan School of Public Health, Boston, Massachusetts, USA).

## Author contributions

**Conceptualization:** Maria Carolina Puça, Tais N. de Sousa, Marcelo U. Ferreira.

**Data curation:** Rodrigo M. Corder, Carlos A. Prete Jr, Igor C. Johansen, Tais N. de Sousa.

**Formal analysis:** Rodrigo M. Corder, Maria Carolina Puça, Carlos A. Prete Jr.

**Funding acquisition:** Rodrigo M. Corder, Tais N. de Sousa, Marcelo U. Ferreira.

**Investigation:** Rodrigo M. Corder, Maria Carolina Puça, Carlos A. Prete Jr, Winni A. Ladeia, Priscila T. Rodrigues, Igor C. Johansen, Tais N. de Sousa, Marcelo U. Ferreira.

**Methodology:** Rodrigo M. Corder, Winni A. Ladeia, Priscila T. Rodrigues, Igor C. Johansen, Tais N. de Sousa, Marcelo U. Ferreira.

**Project administration:** Tais N. de Sousa.

**Resources:** Maria Carolina Puça, Priscila T. Rodrigues.

**Software:** Rodrigo M. Corder, Carlos A. Prete Jr, Tais N. de Sousa.

**Supervision:** Rodrigo M. Corder, Tais N. de Sousa, Marcelo U. Ferreira.

**Validation:** Rodrigo M. Corder, Maria Carolina Puça, Winni A. Ladeia, Igor C. Johansen, Tais N. de Sousa.

**Visualization:** Rodrigo M. Corder, Carlos A. Prete Jr.

**Writing – original draft:** Rodrigo M. Corder, Maria Carolina Puça, Carlos A. Prete Jr, Winni A. Ladeia, Priscila T. Rodrigues, Igor C. Johansen, Tais N. de Sousa, Marcelo U. Ferreira.

**Writing – review & editing:** Rodrigo M. Corder, Maria Carolina Puça, Carlos A. Prete Jr, Winni A. Ladeia, Priscila T. Rodrigues, Igor C. Johansen, Tais N. de Sousa, Marcelo U. Ferreira.

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
