## [Decision Letter · Decision Letter 0]

4 Mar 2026

PNTD-D-25-01961

Reduced cytochrome P-450 (CYP) 2D6 activity and Plasmodium vivax malaria risk in Amazonians: a retrospective, population-based cohort study

Dear Prof. Ferreira,

Thank you for submitting to PLOS Neglected Tropical Diseases.

Two reviews of your interesting manuscript are outstanding. Instead of waiting any longer for them to come in, a major revision decision has been taken. Consideration of the combination of the single review available (see the attached file concerned) and the academic editor’s input (duplicated as an attached file and which in terms of its detail amounts to another review) clearly indicates that the two review reports which are late would not alter this major revision decision. You should therefore at this point go ahead with revising the manuscript. Further changes might or might not be requested later on, when the revised the manuscript is assessed.

We look forward to receiving your revised manuscript.

Kind regards,

Miles B. Markus, Ph.D. Imperial College London

Academic Editor

Abhay Satoskar

Section Editor

Shaden Kamhawi

co-Editor-in-Chief

Paul Brindley

co-Editor-in-Chief

**Additional Editor Comments:**

1. In the reference list, provide the names of up to 6 authors (followed by “et al.” if there are more than 6 authors). Use en dashes for page number ranges, not hyphens. As per the journal’s style, “abbreviate” the second page number in those references where it does not need to appear in full (as has been done in line 463). E.g., in line 382, write “116–24”, as opposed to “116-124”; and in line 387, “595–8” instead of “595-598”. At the end of references where the DOI is given, omit the full stop (period) behind the DOI, as at the end of line 441.

2. “time-to-recurrence” is correctly used adjectivally (meaning that there are hyphens) in lines 131, 258 and 309. But there is lack of uniformity, hyphen-wise, where the expression appears non-adjectivally in lines 195 and 254. There are no hyphens in line 195 (which is in order, because that is correct) but they are present in line 254. Delete the hyphens in line 254.

3. “CYP2D6” sometimes appears in italics and sometimes it doesn’t (mostly). Uniformity is needed. I suggest no italics, partly to avoid having to decide about “P450 (CYP)”. But in the reference list, follow what was in the original publications.

4. Both “Figure” and “Fig.” appear in the manuscript (MS) (and in line 177, “Fig” does not have a full stop/period after it, unlike elsewhere). Introduce uniformity by checking the MS in this respect. Copy the style used in some recent publications in the journal.

5. Line (L) 3 to L 4: Inter alia, primaquine per se not being a “treatment”, reword the text something like: “Primaquine (PQ) is the only widely available drug that eliminates latent hypnozoites, thereby preventing relapses …”.

6. L 4: In view of the fact that it is technically the infection which relapses, not the parasite (Plasmodium vivax), substitute “of Plasmodium vivax malaria” for “of the human malaria parasite Plasmodium vivax”.

7. L 5: Insert a hyphen between “P” and “450”, as has been done elsewhere, for uniformity.

8. L 6: Correct “minimal” to “minimum” (there being a subtle difference in meaning in regard to this sentence).

9. L 33: Substitute “drug that eliminates” for “treatment to eliminate”; and insert a comma at the end of the line, after “hypnozoites”.

10. L 34: Substitute “thus preventing” for “and prevent”.

11. L 34: Substitute “of Plasmodium vivax malaria” for “of the human malaria parasite Plasmodium vivax”. The reason is that, as stated in one of the comments above, Plasmodium vivax per se (i.e., the parasite itself) does not relapse.

12. L 36: Correct “minimal” to “minimum”. The reason is given in comment no. 8, above.

13. L 46: The word “neglected” used to be appropriate for P. vivax but in 2026 is doubtfully applicable, depending on how one views the meaning of “neglected”. A lot of research involving P. vivax is being carried out these days. Therefore, consider deleting “neglected”. But leave it there if you like it.

14. L 47 & L 114, Etc.: Abbreviate “Plasmodium vivax” as “P. vivax” in these two lines. But in line 323 (heading for “S5 Table”) as well as in the headings for Tables 3 and 4, “Plasmodium” should appear in full, as it does in other headings.

15. L 49: Substitute “give rise to” for “originates”. Also in this line, delete “repeated”, because coming after “one or more”, it is saying the same thing twice.

16. L 49: With reference to what one of the reviewers wrote, insert “symptomatic” in front of “malaria” here (and elsewhere in the MS, where appropriate).

17. L 49: Delete “repeated” because coming after “one or more”, it is saying the same thing twice.

18. L 50: It is suggested that the comma behind “episodes” be deleted and that “often occurring” be inserted in front of “several”, because relapses can also take place outside the range given in line 50.

19. L 56: Add an s” to “account” to make it “accounts” (the antecedent in the sentence being “activity”).

20. L 56: After “[5]” in this line, consider inserting a sentence such as: “It has recently been suggested that, because of this mechanism of action [5], the massive, newly discovered, hidden extrahepatic reservoir of asexual P. vivax parasites in the spleen and bone marrow might (also) be targeted by primaquine [REF.]”.

The sole original primaquine-associated reference here is PMID 31522991 (alternatively or additionally, PMID 10223033).

The relevance hereof is that this mode of parasite inactivation, if it does take place to a significant extent (synergistically or otherwise with especially chloroquine; and for the logical biochemical reasons concerned), will obfuscate what the number of relapses is, because recrudescences will also be prevented. But by including a sentence like this, the authors would then be covered in regard to what the MS otherwise states overall, considering that the asexual parasite inhibition matter will have been mentioned in passing (as above). [You referred previously to this splenic parasite accumulation phenomenon in the paper you have self-cited in this MS as reference 16.]

21. L 68 & L 69: Alter “[9-11]” in this line to “[9–11]” and “[12-14]” in the next line to “[12–14]”. In other words, substitute en dashes for the hyphens. An en dash is used for a reference range, not a hyphen.

22. L 97: Check whether you do wish to use italics for “ACKR1”.

23. L 101: “whatever” should be changed to “whichever” because you are referring back to specific options here.

24. L 112: “malaria infection” appears in this line. Although likewise to be found in publications frequently, it is nevertheless a misnomer. “Malaria” is by definition an infection, so to add “infection” after it is amounts to saying the same thing twice. Thus, “infection” behind “malaria” is superfluous. Either delete “malaria” in this line or leave it there and substitute “case” for “infection” (assuming that this is what you mean).

25. L 112: Consider inserting “plasmodial” between “different” and “species”.

26. L 113: Insert “had” after “have” so that the wording becomes “have had”.

27. L 114: Abbreviate “Plasmodium” as “P.”, as requested in comment no. 14.

28. L 115: Consider inserting “those of” after “such as”.

29. L 131: Correct “run” to “ran”.

30. L 132: Insert “episode” between “malaria” and “experienced”.

31. L 133: Insert “drug” between “partner” and “other”.

32. L 134: Insert “malarial” in front of “recurrence” because “P. vivax recurrence” is incorrect (it is not the parasite that recurs, as already pointed out in other comments here).

33. L 137 & L 139: Should “site” in these two lines be “area”? Where the study was carried out seems larger than a “site”.

34. L 137: Consider altering the “is unlikely” assertion to the less definite “seems unlikely”.

35. L 138: Substitute “did not suffer” (or “did not experience”) for “remained free of”, which is not a good expression to use here.

36. L139: Change the en dash in “180–day” (which is not a range, for which an en dash would have been correct) to a hyphen (“180-day”).

37. L 139: Insert “period” behind “follow-up”. The way it currently reads in what one would say in conversation (insufficiently formal for a publication).

38. L 140: Change “whatever” to “whichever”.

39. L 156: Alter “differed in” to “differed as regards”.

40. L. 156: Insert “an” in front of “individual’s”.

41. L 185: Correct “rising” to “raising”.

42. L 190: Substitute “recurrence-free” for “free of recurrence”.

43. L 198: Substitute “following infection with P. vivax” for “following a P. vivax infection”.

44. L 205: Correct the en dash in “person–years” to a hyphen (“person-years”).

45. L 224: Delete “a” in front of “severity” (it is subtly incorrect English to use “a” there).

46. L 228: Insert “of” between “20%” and “high-risk”.

47. L 234: Delete “a” in front of “normal”, to correct the English (“a” does not go together with “activity” in this sentence).

48. L 249: At the beginning of the line, delete “A” in front of “reduced”, to correct the English.

49. L 249: Correct “minimal” at the end of the line to “minimum”.

50. L 251: Correct the sentence by inserting “malaria” after “P. vivax”.

51. L 259: Substitute “that causes” for “causing” (finite verb needed instead of present participle).

52. L 263: Delete “a” in front of “CYP2D6”.

53. L 272 & L 273: You could leave “(less likely)” there if that is what you believe because nobody is about to prove you wrong. However, other possibilities are to, more safely, insert “perhaps” in front of “less”; or, even more safely, to delete “(less likely)”.

54. L 273: It has become apparent that it is highly likely, parasitologically, that both early and late P. vivax recrudescences take place. To cover the authors and also to reflect their awareness, consider inserting PMID 28366603 (key original reference for this probability) after “late recrudescences” in this line. Thus, “… late recrudescences [REF.] collectively defined …”.

55. L 273: Think about whether your paper would be academically enhanced by adding the following recent (2026) citation in this line at the end of the sentence after “recurrences”: PMID 41324557. The publication fits in with your statement about not separating recurrences into 3 categories, by showing how it can be attempted under the right circumstances. Citation of the article would at the same time provide the reader with a topical reference that gives further insight into your definition of “recurrences”. Thus, “… collectively defined as recurrences [REF.]. Relapse frequencies were not …”.

56. L 274: Delete “the” in front of “adherence”. This is an example in English of where omission of “the” is grammatically and logically correct (unlike instances where “the” should indeed appear).

57. L 278: Substitute a full stop for the comma after “1.0”; delete “with” in the line; and start a new sentence by inserting “This has” in front of “practical”.

58. L 283: It looks as though there is a space which should not be there between “Calil” and the comma that follows.

59. L 308: Insert “drug” between “partner” and “other”.

60. L. 309: Substitute “131 first or only P. vivax malaria recurrences” for “131 P. vivax first or only recurrences”.

61. L 310: Insert “period” after “follow-up”. The latter on its own is too informal for a publication such as this one.

62. L 321: Substitute “2014–2018” for “2014-18”, using an en dash, not a hyphen, because it is a range (for which en dashes are used rather than hyphens).

63. L 323: Insert “malaria” after “P. vivax” (before “recurrence”).

64. L 324. Alter “2014-18” as described in the comment for line 321.

65. L 325: Is something missing behind “S1 File”?

66. L 354: Although “Secretaria” might, I dare say, be used for the department or office (“team” in line 353), translation into English would make sense to me as “at the Health Secretariat” (the equivalent in English), rather than “at the Health Secretary” (considering that in English, “Secretary” is a single person). I suggest that you alter “Secretary” to “Secretariat”.

67. L 368 to L 370: Citation of the 2024 World Malaria Report is fine. But should the same information in respect of why you are citing it happen to be in the 2025 report, that would potentially be citable (as a more recent report): https://www.who.int/publications/i/item/9789240117822

68. L 381: For uniformity, use a lower case “c” in “consensus (after the colon), as has been done after colons elsewhere in the reference list.

69. L 385: Alter the hyphen in “1381-2” to an en dash (“1381–2”) because it is a page range in this instance. Actual page numbers are involved, not an article number (as in line 391, for example, where the hyphen is correct). The abbreviated ”1381–2” is correct in this line, being the journal’s style (not the “1381–1382” style used for various other references listed and in respect of which changes by the authors have been requested via the first comment in this editor’s report).

70. L 392: Insert a hyphen between the “S” and “B” after “Ballard” to change from “SB” to “S-B”.

71. L 394: Use an upper case first letter in “higher” so that it becomes “Higher”. [The “A” in front of it is not the beginning of a sentence, but an initial of an author.]

72. L 416: Substitute “Rodrigues PT” for “Rodrigues P” (add “T” as a second initial, which is in the original publication).

73. L 417: Just as you used the italics that were in the title of the original reference 14 paper, put “CYP2D6” in this line in italics.

74. L 418: Substitute “2025;232:e571–9” for “2025 Aug 19:jiaf412”.

75. L 425: Use an upper case “U” in “urban” (as in the original publication).

76. L 433: Extend the initials for author Viana to “GMR”.

77. L 453: Substitute “2025;81:379–86” for “2025:ciae482”.

78. Headings for Table 1, 2, 3, 4 & 5 in main MS: In each table, change “2014-18” to “2014–2018, using an en dash instead of a hyphen.

79. In the heading for Table 2 in the main MS, insert “malaria” between “P. vivax” and “recurrence”.

80. Reminder: Reduce the number of authors’ names given in the reference list in accordance with comment no. 1 here.

MILES B. MARKUS (Editor)

**Journal Requirements:**

At this stage, the following Authors/Authors require contributions: Marcelo U. Ferreira. Please ensure that the full contributions of each author are acknowledged in the "Add/Edit/Remove Authors" section of our submission form.

2) Tables should not be uploaded as individual files. Please remove these files and include the Tables in your manuscript file as editable, cell-based objects. For more information about how to format tables, see our guidelines:

https://journals.plos.org/plosntds/s/tables

3) Some material included in your submission may be copyrighted. According to PLOSu2019s copyright policy, authors who use figures or other material (e.g., graphics, clipart, maps) from another author or copyright holder must demonstrate or obtain permission to publish this material under the Creative Commons Attribution 4.0 International (CC BY 4.0) License used by PLOS journals. Please closely review the details of PLOSu2019s copyright requirements here: PLOS Licenses and Copyright. If you need to request permissions from a copyright holder, you may use PLOS's Copyright Content Permission form.

Potential Copyright Issues:

i) Please confirm (a) that you are the photographer of S1B, or (b) provide written permission from the photographer to publish the photo(s) under our CC BY 4.0 license.

ii) Figure 1A. Please (a) provide a direct link to the base layer of the map (i.e., the country or region border shape) and ensure this is also included in the figure legend; and (b) provide a link to the terms of use / license information for the base layer image or shapefile. We cannot publish proprietary or copyrighted maps (e.g. Google Maps, Mapquest) and the terms of use for your map base layer must be compatible with our CC BY 4.0 license.

4) Please note that your Data Availability Statement is currently missing the DOI/accession number of each dataset OR a direct link to access each dataset. If your manuscript is accepted for publication, you will be asked to provide these details on a very short timeline. We therefore suggest that you provide this information now, though we will not hold up the peer review process if you are unable.

5) In the online submission form, you indicated that  Researchers who are interested in potential collaboration should contact the corresponding author (muferrei@usp.br).. All PLOS journals now require all data underlying the findings described in their manuscript to be freely available to other researchers, either

1. In a public repository

2. Within the manuscript itself

3. Uploaded as supplementary information.

7)  Please ensure that the funders and grant numbers match between the Financial Disclosure field and the Funding Information tab in your submission form. Note that the funders must be provided in the same order in both places as well.

**Reviewers' Comments:**

Reviewer's Responses to Questions

**Key Review Criteria Required for Acceptance?**

**Methods**

-Are the objectives of the study clearly articulated with a clear testable hypothesis stated?

-Is the study design appropriate to address the stated objectives?

-Is the population clearly described and appropriate for the hypothesis being tested?

-Is the sample size sufficient to ensure adequate power to address the hypothesis being tested?

-Were correct statistical analysis used to support conclusions?

-Are there concerns about ethical or regulatory requirements being met?

Reviewer #1: (No Response)

**Results**

-Does the analysis presented match the analysis plan?

-Are the results clearly and completely presented?

-Are the figures (Tables, Images) of sufficient quality for clarity?

Reviewer #1: (No Response)

**Conclusions**

-Are the conclusions supported by the data presented?

-Are the limitations of analysis clearly described?

-Do the authors discuss how these data can be helpful to advance our understanding of the topic under study?

-Is public health relevance addressed?

Reviewer #1: (No Response)

**Editorial and Data Presentation Modifications?**

Reviewer #1: (No Response)

**Summary and General Comments**

Reviewer #1: See attachment

PLOS authors have the option to publish the peer review history of their article (what does this mean? ). If published, this will include your full peer review and any attached files.). If published, this will include your full peer review and any attached files.

**Do you want your identity to be public for this peer review?**  For information about this choice, including consent withdrawal, please see our  For information about this choice, including consent withdrawal, please see our Privacy Policy ..

Reviewer #1: No

**Figure resubmission:**
---

## [Editor Report · Decision Letter 1]

18 Mar 2026

Dear Authors,

We are pleased to inform you that your manuscript 'Reduced cytochrome P-450 (CYP) 2D6 activity and Plasmodium vivax malaria risk in Amazonians: a retrospective, population-based cohort study' has been provisionally accepted for publication in PLOS Neglected Tropical Diseases.

PLEASE NOTE, BELOW, THE EDITOR'S LIST OF MINOR MATTERS REQUIRING ATTENTION FROM YOU AT THE PROOF STAGE (or before, should the journal ask you to deal with a manuscript version before then for journal formatting-related reasons). The academic editor's list below is duplicated as an attached file.

Before your manuscript can be formally accepted, you might need to carry out some formatting changes, which you would receive in a follow-up email. A member of our team would be in touch with a set of requests.

Please note that your manuscript will not be scheduled for publication until you have made any journal requirement changes that might be needed, so a swift response would be appreciated.

Kind regards,

Miles B. Markus, Ph.D. (Imperial College London)

Academic Editor

Abhay Satoskar

Section Editor

Shaden Kamhawi

co-Editor-in-Chief

Paul Brindley

co-Editor-in-Chief

1. Line (L) 60: Substitute “PQ” for “primaquine”.

2. L 89 & L 105 & L 339 & L 349: Change “site” to “area” in all four places. And check the supplementary documents in this respect.

3. L 96: Change “1,862 had DNA samples available” to “DNA samples were available for 1,862”.

4. L 102: Consider inserting a full stop (period) after “surface” in this long sentence. Then start a new sentence by inserting “This is” in front of “because” in that line.

5. L 103: Consider inserting a comma after “[20]” and substituting “which” for the word “and” that currently appears behind “[20]”.

6. L 104: Consider inserting “the” between “for” and “ACKR1”.

7. L 106: Consider altering “they lacked” to “we lacked”.

8. L 108: Insert the apparently missing closing bracket after “mutation”.

9. L 108: Consider substituting “remained” for “were retained”.

10. L 110: “counted” could potentially be substituted for “enumerated” as a change from the “enumerated” that appears in line 92. Otherwise, leave this as “enumerated”.

11. L 122: Insert “to” between “equal” and “or”.

12. L 147: At the end of the line, delete the “a” in front of “P. vivax”.

13. L 157 & L 158: Check the bracketing in these two lines. Is a closing bracket missing?

14. L 232: Does “risk” fit in there in this particular instance (read the sentence carefully)? Should it be deleted or not? If so, should it simply be deleted or replaced by “possibility”?

15. L 248: Change “proxy of” to “proxy for”.

16. L 316: Substitute “varying” for “variable”.

17. L 350: Delete “they lacked” and insert “was not available” at the end of the line, after “information”.

18. L 351: At the end of the line, substitute “remained” for “were retained”.

19. L 353: The style in which dates are set out here (e.g., “01 January 2014”) is not the same as in line 361, for example, where the date appears as “January 1, 2014”. I happen to prefer the former style. Whether or not the journal uses a particular style aside, introduce uniformity in the whole manuscript.

20. L 353: Correct “2028” to “2018”.

21. L 361: Insert the word “and” between the dates, deleting the comma after “2014”.

22. L 361: Correct “2028” to “2018”.

23. L 376: Alter “variable” to “variables”.

24. L 384: Insert a comma after “resources”.

25. L 417 to 419: “We thank the Program …” appears here in the “Financial disclosure” section. That sentence should be moved to the “Acknowledgements” section.

26. L 428: Change “979–990” to “979–90”.

27. L 463: Italicize “Plasmodium vivax”.

28. L 508: Delete the space after “86” as well as “ciae482” so that the full stop after it comes immediately behind “86” (i.e., “… 86.”).

29. L 519: At the end of the line, use a lower case “a” in “an” after “women:”. In other words, change “An” to “an”.

30. L 523: Remove the space between “2026” and the semicolon after it (close up the gap).

MILES B. MARKUS (Academic Editor)

---

## [Editor Report · Acceptance letter]

Dear Prof. Ferreira,

We are delighted to inform you that your manuscript, "Reduced cytochrome P-450 (CYP) 2D6 activity and Plasmodium vivax malaria risk in Amazonians: a retrospective, population-based cohort study," has been formally accepted for publication in PLOS Neglected Tropical Diseases.

Best regards,

Shaden Kamhawi

co-Editor-in-Chief

Paul Brindley

co-Editor-in-Chief
